# Study on Key Parameters for Jet Impacting Pulverized Coal Deposited in Coal-Bed Methane Wells

Hongying Zhu [1],*, Likun Xue [1], Fenna Zhang [1],*, Yaoguang Qi [1], Junwei Zhao [2] and Dehua Feng [1]

[1] College of Mechanical and Electronic Engineering, China University of Petroleum (East China), Qingdao 266580, China
[2] Tianjin Branch of CNOOC Ltd., Tianjin 300459, China
* Correspondence: zhuhongying@s.upc.edu.cn (H.Z.); 20150069@upc.edu.cn (F.Z.)

**Abstract:** Cleaning out the pulverized coal deposited at the bottom of a coalbed methane (CBM) well is key to achieving continuous CBM drainage and prolonging the workover period. In this study, Fluent is used in conjunction with the standard k-$\varepsilon$ model and the Eulerian-Eulerian model to simulate and analyse jet erosion of deposited pulverized coal particles. The depth and width of the stable erosion pit that is formed by jet-impacting deposited pulverized coal under different conditions are determined and provide a theoretical basis for the cleanout of pulverized coal in the bottom of a CBM well. In this paper, the three parameters of the jet target distance, nozzle diameter and nozzle outlet flow velocity are selected to perform an orthogonal simulation. The change trends in the depth and width of the scouring pit with time are determined. The results show that jet impacting of deposited pulverized coal can be categorised into four stages, periods of rapid growth, stability, jet swing and dynamic stability. A sensitivity analysis shows that the nozzle outlet flow velocity has the strongest influence on the depth of the scouring pit among the selected parameters. The depth of the jet impact pit can reach the maximum depth at t = 3 s, while the width of the impact pit can reach the maximum after t = 7 s. This can provide key design parameters for CBM well pulverized coal impacting operation. It is of great significance for capacity damage control during CBM well workover operation.

**Keywords:** coal-bed methane well; pulverized coal; submerged water jet; CFD simulation





## 1. Introduction

Pulverized coal is inevitably generated during the process of coalbed methane (CBM) drainage [1,2]. CBM well production is a process of drainage and gas production [2–4] during which generated pulverized coal particles enter the wellbore with flowing water [5]. Timely discharge of these particles from the wellbore is required to prevent well failures, such as stuck and buried pumps, which significantly affect the continuous drainage and gas production efficiency of CBM and reduce economic benefits [6–8]. The discharging of pulverized coal deposited at the bottom of CBM wells has become an important factor for CBM well workover operations.

Considerable research has been done on how jet erosion affects particle sedimentation. Gong simulated the impact of an air jet on coal particles, determined the trend of the depth and width of the pit with the jet pressure, and verified the results experimentally [9]. Xia performed simulations to investigate the effect of the submerged depth on the jet erosion of non-adhesive deposited sediment: changes in the water depth had no discernible effect on scouring, whereas reducing the water depth decreased the development rate of the pit depth [10]. Huai used a two-dimensional RNG $k$-$\varepsilon$ turbulence model (Renormalization Group Theory. A mathematical theory which can be used to derive a turbulence model which is similar to the standard $k$-$\varepsilon$ model) and dynamic grid technology to simulate the erosion of a sandy riverbed by the action of a vertical jet and identified three main stages

of erosion: rapid growth, stability and jet swing [11]. Zhao used the RANS (Reynolds-averaged Navier-Stokes equations. A series of time-averaged equations of motion for fluid flow) equation and suspended sediment transport equation to study the local erosion of piggyback pipelines [12]. Jin carried out simulations to determine changes in various factors under multiple conditions and verified the results through tests [13].

In recent years, much attention has been focused on the output mechanism of pulverized coal [14,15], the pattern of motion of pulverized coal in a coal seam [5,7], and the influence of the drainage work system on the motion of pulverized coal [16,17].

In terms of the output mechanism and the pattern of motion of pulverized coal in a coal seam, Liu & Wei analysed the effect of concentration, pulverized coal size, composition, and mechanism and control measures of pulverized coal in the coal seam, Qinshui Basin, China [6,18]. Bai numerically characterized the generation process of pulverized coal at the micro-scale, Sydney Basin & Bowen Basin, Australia [19,20]. Han & Zhang analysed the migration of pulverized coal in the coal seam during production and determined the critical transport velocity to the terminal settling velocity in the coal seam, Qinshui Basin, China [15,21]. The output and migration of pulverized coal lead to permeability damage in the coal seam during the dewatering period [22,23].

However, the pulverized coal entering the wellbore cannot be discharged completely from CBM wells. Continuously increasing quantities of deposited pulverized coal will cause production problems, such as buried pumps and stuck pumps [24]. At present, hollow pumping rod flushing or jet pump flushing is mainly used to address the problem of deposited pulverized coal [2]. Pulverized coal contains viscous substances, such as coal coke [21,24]. The mechanism of pulverized coal flushing and the factors affecting the process of pulverized coal flushing are not clear, which has limited the efficiency of pulverized-coal flushing.

In this paper, considering the effect of coal coke on deposited pulverized coal, we simulate the deposited pulverized coal flushing process by jet stream with Computational Fluid Dynamics (CFD) method. The analysis of the flow pattern with time was studied. Meanwhile, we analysed the key factors affecting the deposited pulverized coal flushing process. Furthermore, this work can provide the theoretical basis for future research on special pulverized coal flushing devices or flushing processes for CBM wells. It can further improve the efficiency of flushing pulverized coal in CBM wells and promote the development and exploitation of CBM resources.

## 2. Theoretical Model

The submerged jet technique is used to remove pulverized coal that has been deposited in a wellbore. A power liquid is used to scour solid deposited particles. After the start of the jet, the flow in the area is complicated due to the contact and mixing between the jet and well fluid, and the interaction between the jet and the coal powder particles. It is difficult to analyse by theoretical method, and it is necessary to discretize this domain and simulate by CFD method. The flow regimes include laminar, turbulent, liquid-solid two-phase flow, etc. A theoretical basis for performing calculations on these various flow patterns is given below.

Equation (1) is the mass conservation equation for the pulverized coal and dynamic liquid phases [25]:

$$\begin{cases} \frac{\partial[\rho_w(1-S_c)]}{\partial t} + \nabla[\rho_w(1-S_c)\boldsymbol{V}_w] = 0 \\ \frac{\partial(\rho_c S_c)}{\partial t} + \nabla(\rho_c S_c \boldsymbol{V}_c) = 0 \end{cases} \tag{1}$$

where $S_c$ is the volume fraction of the pulverized coal phase deposited at the well bottom; t is time; $\rho_c$ is the density of the pulverized coal and $\rho_w$ is the density of the dynamic liquid; $\boldsymbol{V}_c$ is the transport velocity vector of the pulverized coal and $\boldsymbol{V}_w$ is the transport velocity vector of the dynamic liquid.

Equation (2) is the momentum conservation equation for the pulverized coal and power liquid-liquid phases [26]:

$$\begin{cases} \frac{\partial[\rho_w(1-S_c)V_w]}{\partial t} + \nabla[\rho_w(1-S_c)V_wV_w] \\ = div[(1-S_c)\sigma_w] + \rho_w(1-S_c)m + F_e + F_w \\ \frac{\partial(\rho_cS_cV_c)}{\partial t} + \nabla(\rho_cS_cV_cV_c) = div(S_c\sigma_c) + \rho_cS_cm + F_e + F_c \end{cases} \tag{2}$$

where $F_e$ is the bonding force between the pulverized coal particles, $F_c$ is the bonding force between the pulverized coal particles, and $F_w$ is the additional force between the pulverized coal phase and the power liquid-liquid phase, respectively, with $F_c = -F_w$; and $m$ is the gravitational force per unit volume of the pulverized coal phase or the power liquid-liquid phase. $c$ is the stress tensor of the pulverized coal phase and $w$ is the stress tensor of the power liquid-liquid phase.

The counter diffusion of the pulverized coal phase is relatively small and can therefore be neglected; the fluctuating energy equation of the pulverized coal phase is then obtained as Equation (3) [27]:

$$\frac{1.5\rho_c dv_c^2}{dt} = S_c : \nabla V_c - \varepsilon \tag{3}$$

where $\varepsilon$ is the energy loss rate due to inelastic collisions between pulverized coal particles, and $S_c$ is the partial stress tensor of the pulverized coal phase.

The intrinsic equations of the power fluid-liquid phase and the pulverized coal phase at the bottom of the wellbore are given by Equations (4) and (5) [28]:

$$\sigma_c = -p_c n + S_c \tag{4}$$

$$\sigma_w = -p_w n + S_w \tag{5}$$

where $S_w$ is the partial stress tensor of the power liquid-liquid phase, $n$ is the unit tensor, $p_c$ is the pressure at a prescribed point in the pulverized coal and $p_w$ is the pressure at a prescribed point in the pulverized coal and power liquid phase.

## 3. Modelling and Meshing

### 3.1. Parameters of the Model

A CBM well in the Linxing block of the Ordos Basin of China is selected. The wellbore parameters are as follows: the outer diameter of the casing is 140 mm and inner diameter of the casing is 121 mm. Figure 1 shows the two-dimensional geometric model established using Gambit, where the model has a length of 121 mm and a height of 200 mm. The nozzle has a length of 10 mm and a diameter $d$. the height of the pulverized coal area is $h$, and the distance between the nozzle outlet to the upper surface of the pulverized coal is $H$. The calculation domain is divided into rectangular cells.

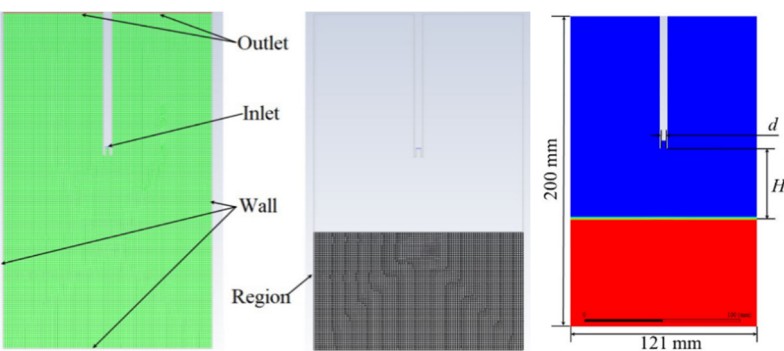

**Figure 1.** Geometric models and meshing for CFD.

The initial parameters of the coal seam are as follows. The temperature of the coal seam is 292 K. The density of pulverized coal particles is 1490 kg/m$^3$. The size of the pulverized coal particle is 0.0018 mm and the viscosity between pulverized coal particles is 0.15 Pa·s. The density of the fluid is 1010 kg/m$^3$ and the viscosity of the fluid is 0.88 mPa·s. Orthogonal design is used to select the parameters such as the target distance, nozzle diameter, and nozzle outlet flow rate. The parameter values are shown in Table 1.

**Table 1.** Parameters for orthogonal design CFD simulation.

| No. | $H$ **(mm)** | $d$ **(mm)** | $V$ **(m/s)** |
|---|---|---|---|
| 1 | 28.6 | 2 | 0.5 |
| 2 | 38.7 | 3 | 1.0 |
| 3 | 63.8 | 4 | 1.5 |

### 3.2. Verification of This Case

The verification of the independence of the spatial grid size and the time-step was carried out, and the related calculation results are shown in Figure 2. With the spatial grid size independence verification, when the number of grids in the computational domain is greater than 41,688, the maximum change of flow velocity of the jet at 0.5 s is less than 0.1%. Therefore, the number of domain grids is set to 41,688. With the time-step independence verification, when the iteration time interval is 0.001 s and $5 \times 10^{-5}$ s, the maximum change of the jet velocity at 0.5 s is less than 0.2%. Considering the computational cost and time, the iteration time interval is set to 0.001 s.

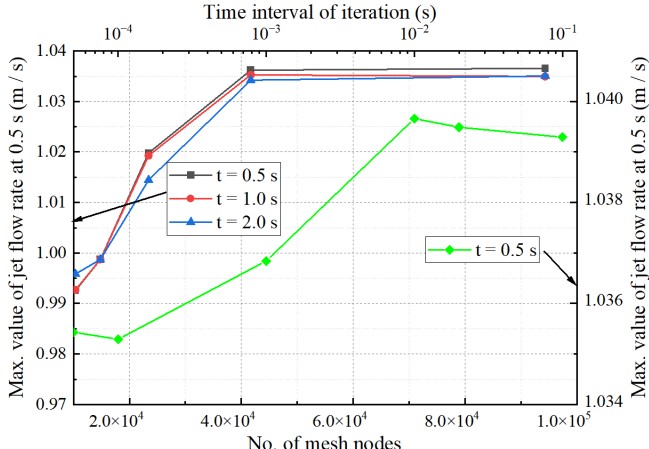

**Figure 2.** Independence Verification for CFD.

## 4. Discussion

A jet flushing device is lowered to the bottom of a CBM wellbore and used to flush out deposited pulverized coal. FLUENT software (version 2019 R1) is used to carry out the simulation. Considering the complex flow field in the domain, the multiphase Eulerian mode is selected. The Eulerian phase is selected as solid-liquid. The standard $k$-$\varepsilon$ model is applied in turbulence mode. The basic parameters are set to the physical properties of the deposited pulverized coal. The simulation is performed using the coupled method with a target distance $H$ of 28.65 mm, a nozzle diameter of 1.5 mm, and a nozzle outlet flow rate of 1.0 m/s. Figure 3 shows the phase cloud diagram of pulverized coal determined by the flushing simulation. Under the condition of constant target distance and jet velocity, the upper profile of the deposited pulverized coal changes with time when the jet flushes the coal powder. When t = 2 s, the depth of the impact crater reaches the maximum. The jet deflects to the left at 4 s, and the jet deflects to the right at 7 s. The width of the impact crater increases with the left and right deflection of the jet. The depth and width of the impacting pit are defined in Figure 4.

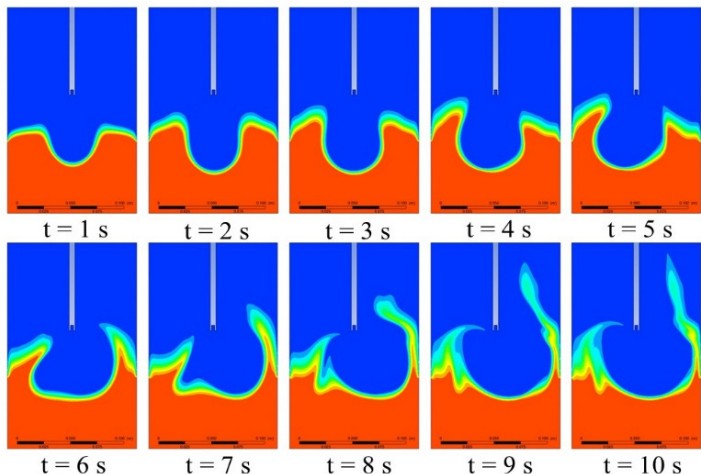

**Figure 3.** Cloud diagram for the pulverized coal phase.

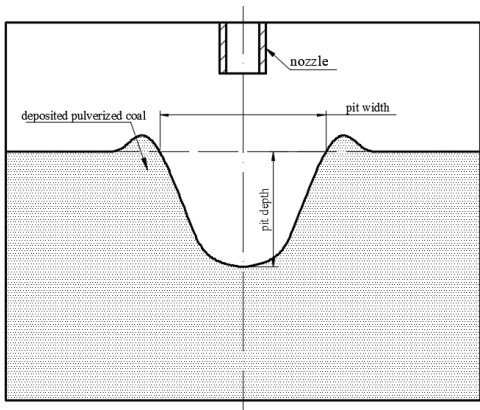

**Figure 4.** Jet scouring pit depth and pit width.

Three sets of 10-s simulations are conducted for different *H*, *d* and *v*. Figure 5 shows the trends for the depth and width of the scouring pit formed by the submerged jet under different conditions.

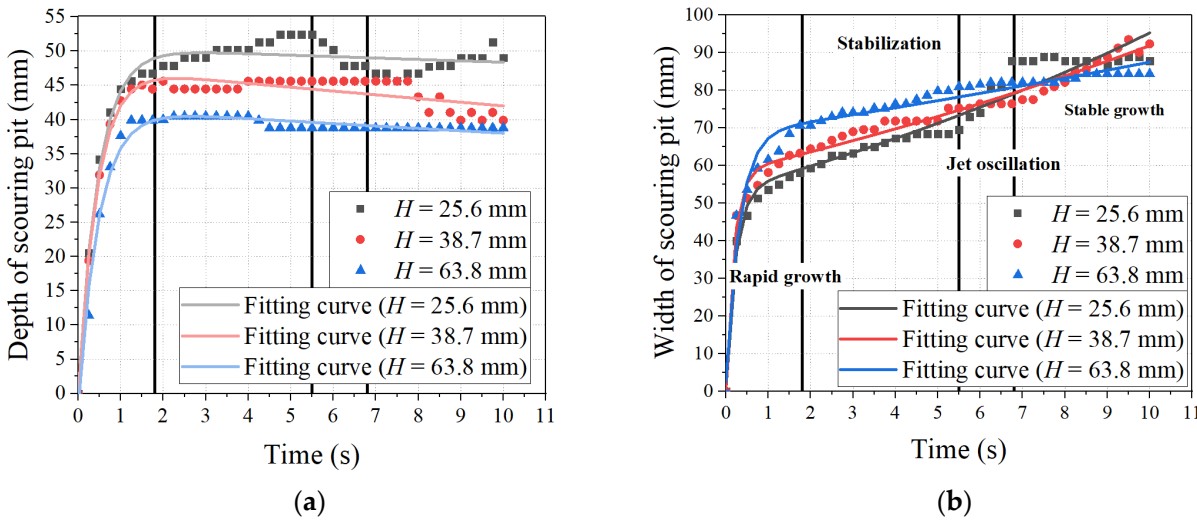

(a)                                                          (b)

**Figure 5.** Temporal variation corresponding to different target distances for the (**a**) depth and (**b**) width of the scour pit.

Figure 5 shows that the formation of scouring pits by impacting the deposited pulverized coal with the jet can be categorised into four stages.

(a) The depth and width of the pit grow rapidly during the first stage. At this stage, the jet has left-right symmetry around the jet axis.

(b) The second stage is the stabilization period. The scouring pit stabilizes at a well-defined depth and width. At this stage, the scouring pit may also continue to slightly grow in size.

(c) The jet exhibits oscillations during the third stage. The jet becomes asymmetrical and starts to "oscillate" from left to right, where the inner surface of the casing (system boundary) is the stationary point of the oscillation. The jet continues to scour the left and right sides of the scour pit, which continues to expand in width but not in depth.

(d) The fourth stage is the dynamic stabilisation stage. At this stage, the jet continues to oscillate around the jet axis, albeit with a well-defined amplitude, and the scour range is also well-defined. Therefore, the depth and width of the pit do not change significantly, and the scouring pit reaches a state of dynamic equilibrium.

### 4.1. Target Distance

Figure 5 shows the variation in the depth and width of the scour pit with time for different target distances, a nozzle diameter of 3 mm and a nozzle outlet flow rate of 1 m/s. An analysis of the abovementioned results shows that the larger the jet target distance is, the longer the distance through which the water jet from the nozzle moves before reaching the surface of the deposited pulverized coal. The submerged environment presents viscous resistance to the liquid jet ejected from the nozzle. There is a violent energy exchange between the jet and the submerged environment. Therefore, the longer the distance through which the jet moves, the more work is done by the corresponding viscous resistance, and the more energy is lost. Therefore, the speed of the liquid-phase jet reaching the deposited coal particles is reduced accordingly. Ultimately, the scouring effect will be stronger.

Figure 5a shows that before the jet enters the oscillation phase, the pit width also decreases linearly with increasing target distance. After the jet enters the oscillation stage, the pit width continues to increase as the jet oscillates from left to right and scours the inner walls of both sides of the pit. However, the onset time of the jet oscillation stage changes with the target distance. Figure 5b shows that the larger the target distance is, the shorter the onset time of the jet oscillation stage.

The axial distance that the jet needs to pass through to flush out pulverized coal increases with the target distance. As it is difficult to control the energy exchange between the jet fluid and the surrounding submerged environment, the action of the viscous drag force is not symmetrical. Increasing the target distance of the jet also increases the possibility of an uneven energy exchange between the left and right sides. Therefore, the larger the target distance is, the earlier the jet starts to oscillate.

### 4.2. Nozzle Diameter

The following key parameters are used to obtain the simulation results presented in Figure 6: a target distance of 25.6 mm and a nozzle outflow rate of 1 m/s. Figure 6a shows that the scouring pit depth also increases with the nozzle diameter. The larger the nozzle diameter, the higher the outflow rate. Increasing the nozzle diameter results in an increase in the kinetic energy of the fluid flowing out of the nozzle and thereby the depth of the crater generated by jet scouring.

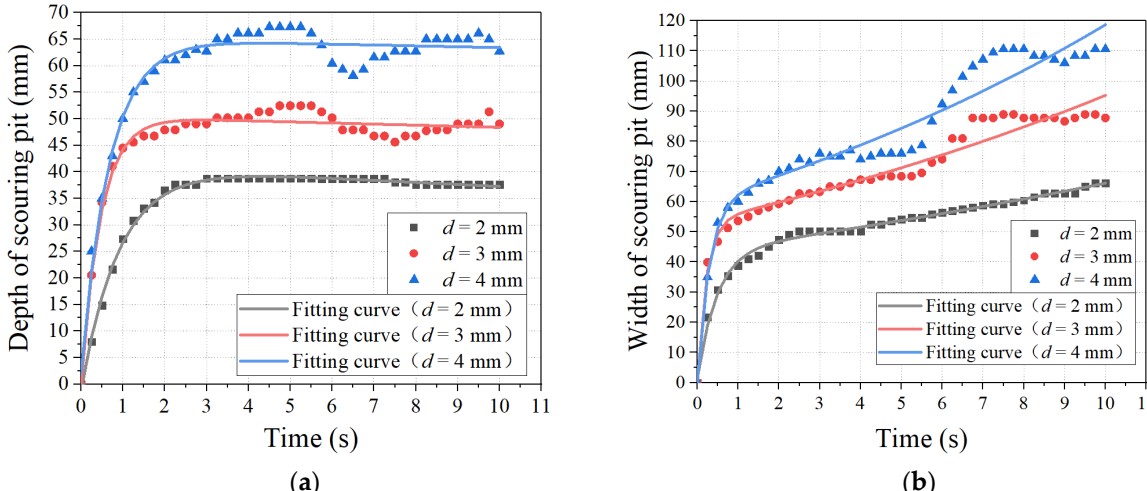

**Figure 6.** Temporal variation corresponding to different nozzle diameters for the (**a**) depth and (**b**) width of the scouring pit.

Figure 6a shows that as the nozzle diameter $d$ increases, the onset time of the jet swing period increases. This result is obtained for the following reason: the larger the nozzle diameter is, the higher the energy of the jet flow is, which increases the period of intense energy exchange between the jet and the submerged environment around the nozzle. However, the loss of fluid energy on both sides of the jet axis is not completely symmetric, inducing oscillations in the jet.

Figure 6b shows that as the nozzle diameter increases, the jet flush crater widens. At a fixed target distance, jet oscillations occur after 5.0 s for all nozzle diameters, where the larger the nozzle diameter is, the more prominent the pit widening produced by the jet oscillation is. For a nozzle diameter of 4 mm, the pit width increases from 76.0 mm (t = 5 s) to 110.6 mm (t = 10 s), corresponding to a 45.5% increase. For a nozzle diameter of 2 mm, the pit width increases from 54.1 mm (t = 5 s) to 66.1 mm (t = 10 s), corresponding to a 22.1% increase.

### 4.3. Nozzle Outflow Velocity

The following key parameters are used to obtain the simulation result presented in Figure 7: a target distance of 28.5 mm and a nozzle diameter of 3 mm. Figure 7a shows that the depth of the scouring pit increases with the nozzle outflow velocity.

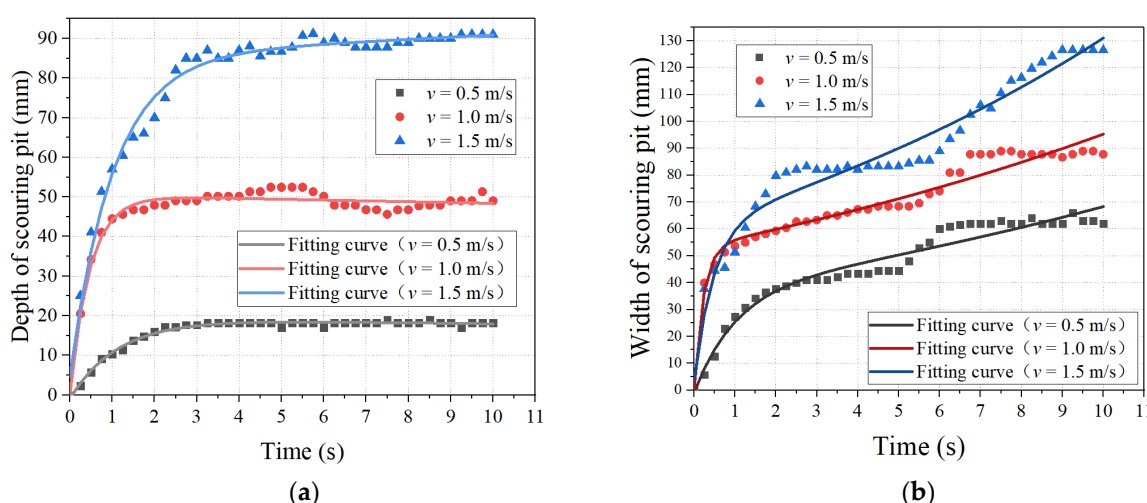

**Figure 7.** Temporal variation corresponding to different nozzle outflow velocities for the (**a**) depth and (**b**) and width of the scouring pit.

Figure 7b shows that as the nozzle outflow velocity increases, the stable scouring pit width formed by the jet before the jet swing period also increases. Figure 7 shows that the higher the nozzle outflow velocity, the longer the onset time for the jet oscillation stage, where the scouring pit width continues to increase during the fourth stage after the jet reaches dynamic equilibrium.

Figure 5a shows that increasing the target distance from 25.6 mm to 38.7 mm results in an increase in the scouring pit depth from 49.3 mm to 43.2 mm (a 51.7% increase in the target distance produces a 12.24% decrease in the scouring pit depth). Increasing the nozzle diameter from 3 mm to 4 mm causes the scouring pit depth to increase from 46.3 mm to 64.4 mm (a 33.3% increase in the nozzle diameter results in a 39.1% increase in the scouring pit depth), as shown in Figure 6a. Increasing the nozzle outlet flow rate from 1 m/s to 1.5 m/s results in an increase in the scouring pit depth from 49.2 mm to 89.3 mm (increasing the nozzle outlet flow rate by 50% results in an 81.5% increase in the scouring pit depth), as shown in Figure 7a. The analysis presented above leads to the conclusion that the nozzle outflow velocity has the most significant effect on the jet scouring pit depth among the three parameters.

Secondary scouring from jet oscillation near the stationary point is a significant influence factor for the scouring pit width. The higher the jet oscillation amplitude is, the broader the scouring range of the jet is. Therefore, a larger target distance increases the oscillation range because of a longer jet axis. It can be concluded that the target distance *H* has the most significant effect on the pit width among the investigated parameters, as shown in Figure 7b.

### 5. Conclusions

The following results were obtained for simulations performed using Fluent for the process of jet impacting pulverized coal at the bottom of the CBM wellbore.

(1) The deposition of pulverized coal by jet impacting can be categorised into four phases: rapid growth, stabilisation, jet oscillation and dynamic stabilisation. It takes about 6.8 s from the start of the jet to the dynamic stabilization stage, which is a guiding role for the residence time of the pulverized coal impacting unit during CBM well impacting. At the same time, there is a decreasing trend of impact depth variation after 6.8 s. Therefore, the residence time of the jet device should be no more than 6.8 s when impacting the pulverized coal at the bottom of the well, at which time the jet flushing device should continue to move downward slowly, in order to ensure the efficiency of impacting pulverized coal.

(2) The onset time for the jet oscillation stage decreases with increasing target distance and increases with the diameter and outflow velocity of the nozzle. There is a constant energy exchange between the jet and the surrounding fluid and solids. The energy loss on both sides of the jet appears to be unequal. When the energy difference between the two sides of the jet develops to more than a certain amount, which varies under different conditions, the phenomenon of jet oscillation occurs.

(3) In the process of jet impacting deposited pulverized coal, the impact pit can reach the maximum depth at 3 s, while it takes 7 s to reach the maximum width. Considering the best flushing effect, it is recommended that the single point hovering time of the equipment should not be less than 7 s during the operation of deposited pulverized coal flushing. If the hovering time is less than 7 s, the jet will not be able to effectively drive the movement of the pulverized coal deposited near the inner wall of the casing. This will significantly reduce the pulverized coal cleaning efficiency.

(4) The depth and width of the pit decrease with increasing target distance. The depth and width of the pit increase with both the diameter and outflow velocity of the nozzle. Therefore, for the parameter design of the jet device, larger nozzles should be selected or the pressure of the power fluid for impacting pulverized coal should be higher while considering the balance between operating costs and efficiency of deposited pulverized coal flushing.

**Author Contributions:** Conceptualization, L.X. and H.Z.; methodology, H.Z.; software, L.X.; validation, F.Z. and J.Z.; formal analysis, F.Z. and D.F.; investigation, H.Z.; resources, Y.Q.; data curation, J.Z.; writing—original draft preparation, L.X. and H.Z.; writing—review and editing, H.Z.; visualization, H.Z. and D.F.; supervision, Y.Q.; project administration, F.Z.; funding acquisition, F.Z. All authors have read and agreed to the published version of the manuscript.

**Funding:** This study was co-supported by the Natural Science Foundation of Shandong Province (ZR2020MD038) and the National Science and Technology Major Special Projects (2016ZX05066004-002 & 2017ZX05064004).

**Institutional Review Board Statement:** Not applicable.

**Informed Consent Statement:** Not applicable.

**Data Availability Statement:** The data that support the findings of this study are available from the corresponding author upon reasonable request.

**Conflicts of Interest:** The authors declare no conflict of interest.

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
