# Peer review of "Study on Key Parameters for Jet Impacting Pulverized Coal Deposited in Coal-Bed Methane Wells"

_coatings, doi:10.3390/coatings12101454_

Round 1
Reviewer 1 Report
Manuscript Number: Coatings-1874831
There are some critical comments that should be resolved, and we can see the revised manuscript again.
1- The paper title needs to be modified to clarify the actual study done.
2- The English language needs to be improved throughout the manuscript.
3- The abstract and conclusion are not clear, so they need to be rewritten to highlight the main findings and recommendations for commercial application.
4- There is no Highlight or nomenclature, so they need to add to clarify the main findings.
5- The manuscript title needs to be modified to clarify the work done.
6- There is no validation for numerical modeling and experimental data.
7- Could you add more details for the simulation section? Also, its layout needs to clarify.
8- How did you choose the simulation condition/program? Are there any limitations in their selections? Please explain in detail?
9- Could you add thermal gravimetric analysis to compare raw oil with produced bio-oil and biodiesel.
10- Could you tabulate the current results with the relevant published studies to highlight the contribution?
11- What is the future investigation needed to relate to this line of work? Please, could you add recommendations/research gaps needed relating to this study?
12- The literature review section needs to rewrite again by updating the reference—some of them as follows.
- https://doi.org/10.1016/j.fuel.2020.119324
-https://doi.org/10.1016/j.csite.2021.101100
Reviewer 2 Report
·
1) In the Abstract the Authors should add some the most important results (its exact values) of performed research. Overall conclusions are well presented, but at least some exact values should be added.
2) I would recommend that all symbols in the text, such as RNG (Line 39) and RANS (Line 42), be clarified. This would make it easier for the reader to understand the information provided in the article.
3) The introduction section is rather less for a paper of this repute. It is highly suggested that the author include more articles on the relevant area.
4) [Line 49]: Line 39: Reference to the literature is missing – “Many useful results…”
5) The novelty section in the present form is not adequate for explaining the work undertaken by the authors. It is suggested that the author explain further of the major methodologies included in the present study in this section.
6) It would be necessary to inform the reader about what formulas (1) and (2) are needed. For now, they are simply placed in the text without additional explanations. In general, it would be necessary also to characterize the model in more detail.
7) [Line 133]: An analysis of Figure 3 would be recommended.
8) It would be advisable to supplement the conclusions with some numerical information obtained from the research.
9) It is necessary to read the article thoroughly and make corrections in English.
Round 2
Reviewer 2 Report
The suggestions have been taken into account. There is no objection to the publication of the article.